# Palladium-catalyzed Suzuki-Miyaura cross-couplings of stable glycal boronates for robust synthesis of C-1 glycals

Anrong Chen[1], Yang Han[1], Rongfeng Wu[2], Bo Yang[1], Lijuan Zhu [3] ✉ & Feng Zhu [1] ✉

C-1 Glycals serve as pivotal intermediates in synthesizing diverse C-glycosyl compounds and natural products, necessitating the development of concise, efficient and user-friendly methods to obtain C-1 glycosides is essential. The Suzuki-Miyaura cross-coupling of glycal boronates is notable for its reliability and non-toxic nature, but glycal donor stability remains a challenge. Herein, we achieve a significant breakthrough by developing stable glycal boronates, effectively overcoming the stability issue in glycal-based Suzuki–Miyaura coupling. Leveraging the balanced reactivity and stability of our glycal boronates, we establish a robust palladium-catalyzed glycal-based Suzuki-Miyaura reaction, facilitating the formation of various C($sp^2$)-C($sp$), C($sp^2$)-C($sp^2$), and C($sp^2$)-C($sp^3$) bonds under mild conditions. Notably, we expand upon this achievement by developing the DNA-compatible glycal-based cross-coupling reaction to synthesize various glycal-DNA conjugates. With its excellent reaction reactivity, stability, generality, and ease of handling, the method holds promise for widespread appication in the preparation of C-glycosyl compounds and natural products.

C-Glycosides, characterized by a carbohydrate moiety connected to an aglycone through a C−C bond linkage, represent critical motifs embedded in bioactive natural products, drug molecules, and glycosylated proteins of eukaryotic cells[1,2]. In contrast to O-linked glycosides, which are susceptible to enzymatic and chemical hydrolysis at the C-O glycosidic bond, C-glycosides offer enhanced chemical and metabolic stability. This feature renders them highly desirable therapeutic agents, particularly as robust artificial surrogates or mimics of native O-glycosides[3]. This concept is exemplified by the development of potent sodium-glucose cotransporter-2 (SGLT-2) inhibitors inspired by the O-glycoside natural product phlorizin. Dapagliflzon (**1**), one such inhibitor, has emerged as blockbuster anti-diabetic drug molecules[4]. Additionally, various C-glycosides like catechin (**2**)[5], kendomycin (**3**)[6], bergenin (**4**)[7], urdamycin B (**5**)[8], and papulacandin D (**6**)[9],

sourced from natural origins, exhibit significant biological activities (Fig. 1a). Consequently, significant strides have been made in recent decades towards constructing C-glycosidic linkages[1,2,10–15]. As part of a diversity-oriented synthesis project, there is a growing demand for versatile and efficient methods to effectively obtain a wide array of C-glycosides.

C-1 glycals, featuring a carbon substituent attached to the C1 center, not only constitute a subclass of C-glycosides but also serve as pivotal synthons for constructing a diverse range of sugar moieties[16]. The presence of the unsaturated enol-ether structure offers significant opportunities for subsequent transformations (Fig. 1b). For example, C1-glycals (**7**) have been utilized as precursors for synthesizing natural 2-hydroxy-C-glycosides via hydroboration-oxidation reactions. Stereoselective hydrogenation of C-1 glycals yields

[1]Frontiers Science Center for Transformative Molecules, Center for Chemical Glycobiology, Shanghai Key Laboratory for Molecular Engineering of Chiral Drugs, School of Chemistry and Chemical Engineering, Zhangjiang Institute for Advanced Study, Shanghai Jiao Tong University, Shanghai, PR China. [2]Discovery Chemistry Unit, HitGen Inc., Chengdu, Sichuan, PR China. [3]Institute of Molecular Medicine, Renji Hospital, School of Medicine, Shanghai Jiao Tong University, 160 Pujian Road, Shanghai, PR China. ✉e-mail: lijuanzhu@sjtu.edu.cn; fzchem@sjtu.edu.cn

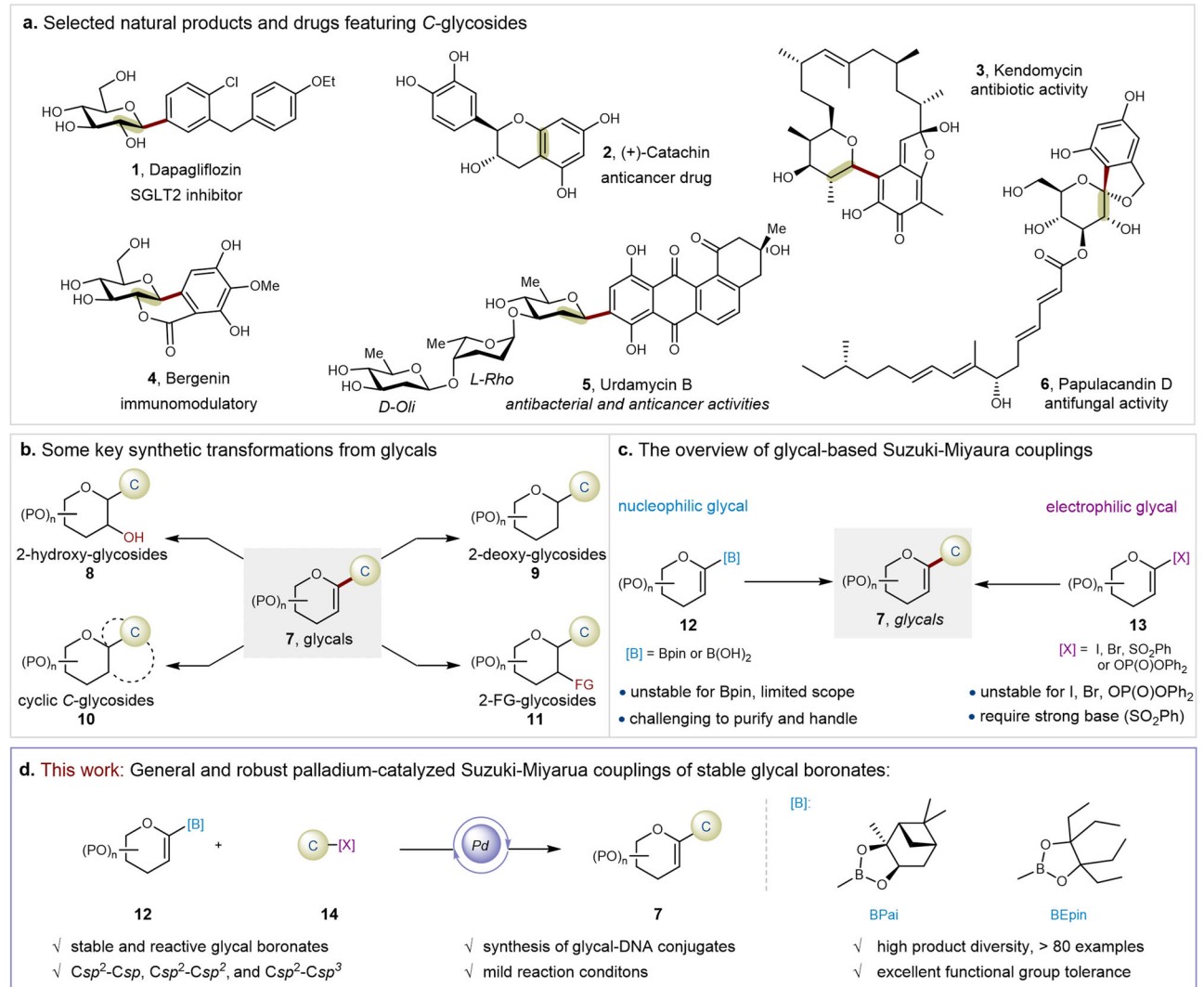

**Fig. 1 | The background and development of glycal-based Suzuki-Miyaura cross-coupling reactions. a** Selected nature products and drugs featuring *C*-glycosides. **b** Some selected synthetic transformations from glycals. **c** The research background of glycal-based Suzuki-Miyaura coupling towards *C*-glycals. **d** Palladium-catalyzed Suzuki-Miyaura coupling of stable glycal boronates (our work). [B]: Borate esters. P: Protecting groups. [X]: Halogens or pseudohalogens. [FG]: Functional groups. C: Carbon electrophiles. Bpai: Pinanediol–boronic acid esters. BEpin: 1,1,2,2-Tetraethylethylene glycol–boronic acid esters.

2-deoxy-*C*-glycosides with excellent anomeric selectivity. Additionally, *C*-1 glycals are widely employed in the preparation of cyclic aryl *C*-glycosides[17]. Regio- and stereoselective *C*2-functionalizations of *C*1-glycals have been easily achieved with excellent yields. As a result, numerous elegant glycal-based coupling approaches have emerged to facilitate access to synthetically useful *C*1-substituted glycals[18–23]. Transition-metal-catalyzed cross-coupling reactions have firmly established themselves as indispensable tools in modern organic synthesis. Notable progress has also been made in constructing *C*1-substituted glycals through transition-metal-catalyzed Stille[24–29], Negishi[30], Hiyama-Denmark[17,31,32], and Heck[33,34] cross-coupling reactions. Among them, the environmentally friendly Suzuki–Miyaura cross-coupling reaction stands out as a crucial tool in glycal-based cross-coupling. Renowned for its reliability, broad availability, and non-toxicity, it facilitates the seamless merging of molecular fragments via C-C bonds, significantly advancing the synthesis of *C*-saccharide mimetics and their diverse applications.

The traditional glycal-based Suzuki–Miyaura cross-couplings involving glycal electrophiles and boron reagents have been extensively studied and have shown significant progress. However, common glycal electrophiles like 1-haloglycals[35] and glycal phosphates[36] often

present several challenges, including instability, easy decomposition, and limited substrate scope. Recently, bench-stable *C*1-sulfonyl glycals were developed as competent glycal electrophiles in Ni-catalyzed Suzuki–Miyaura couplings[37]. Nonetheless, the use of relatively harsh reaction conditions, such as strong base KOH, might restrict their further application in synthesizing complex functional compounds (Fig. 1c). On the other hand, Suzuki–Miyaura cross-couplings of *C*1-glycal boronic acids or their pinacol esters with various electrophiles enjoy broader application and greater interest due to significantly broader substrate scope and dramatically improved coupling efficiencies (Fig. 1c)[38–40]. These advancements have been witnessed in the total synthesis of many natural *C*-glycosides[38,39]. However, manipulating glycal boron species remains a significant challenge. As reported in the literatures[38,41], *C*1-glycal boronic pinacol esters are considered unstable due to the rich electronic properties of their unsaturated enol-ether structure. This instability makes purification difficult, often necessitating special work-up procedures, especially on large scales. Consequently, the challenges related to stability and purification make synthesizing boron glycal species from the corresponding di- or trisaccharides a persistent challenge. Clearly, the utilization of boron glycal species in late-stage modifications of

complex molecules under bioconjugation conditions has not been reported. Therefore, the development of glycal boronates with exceptional stability, reactivity, and ease of handling to achieve efficient and highly modular synthesis of C-1 glycals under mild conditions is of paramount importance.

Organic boron chemistry remains a focal point of sustained attention within the field of organic chemistry[42–47]. Boronic acid protecting groups, continue to emerge, facilitating the development of efficient and versatile transformation reactions[43,48]. Notable examples include N-methyliminodiacetic acid boronates [RB(MIDA)][49], and 1,8-diaminonaphthalene boronamides [RB(dan)][50], known for their remarkable stability under diverse reaction conditions. While RB(dan) and RB(MIDA) find very wide applications, the strongly alkaline conditions required for RB(dan) coupling and the high polarity of RB(MIDA) complicate purification processes. To address these challenges, boronic acid esters derived from 1,1,2,2-tetraethylethylene glycol, known as RB(Epin), have been recently developed[51]. These compounds can be easily purified by chromatography on silica gel, thanks to the strategic spatial protection provided by the four ethyl groups, which dynamically shield the empty orbital of the boron atom. Pinanediol–boronic acid esters [R(Bpai)], leveraging similar stereohindrance, also enhance stability[52,53]. Importantly, this ingenious structural design not only enhances stability but also preserves subsequent C-C cross-coupling activity. Inspired by these exciting advancements, the emergence of C1-glycal boronates with excellent reactivity and stability holds promise in overcoming the current bottlenecks in glycal-based Suzuki-Miyaura coupling reactions.

Herein, we have synthesized two glycal boronates utilizing 1,1,2,2-tetraethylethylene glycol and pinanediol as glycal boronic protecting groups, showcasing excellent reactivity and stability for easy handling. Various glycal boronates are easily prepared from common monosaccharides and disaccharides, isolated in good yields on a gram scale. Leveraging these efficient glycal boronates, we developed a robust palladium-catalyzed Suzuki-Miyaura cross-coupling reaction with diverse (hetero)aryl, alkenyl, and alkyl electrophiles. This methodology enables the efficient assembly of C$sp^2$-C$sp$, C$sp^2$-C$sp^2$, and C$sp^2$-C$sp^3$ bonds in good to excellent yields under mild reaction conditions (Fig. 1d). Our approach has been successfully applied to over 80 examples, including drug molecules, bioactive molecules, oligosaccharides, and oligopeptides. Importantly, thanks to the favorable balance of reactivity and stability exhibited by the glycal boronates, our method has proven to be the efficient example of a DNA-compatible glycal-based cross-coupling reaction, effectively synthesizing an interesting category of carbohydrate-DNA conjugates - glycal-DNA conjugates. In summary, the successful implementation of the aforementioned methods and applications relies on the utilization of interesting glycal boronates characterized by exceptional stability and reactivity.

## Results and discussion
### Reaction development
The cornerstone of developing a general, highly modular, and robust glycal based cross-coupling reaction lies in structurally diverse C1-glycal boronates with excellent reactivity and stability. To achieve this, our investigation delved into synthesizing efficient C1-glycal boronates. Initially, we modified the Ir-catalyzed C1-selective C – H borylation of glycal derivatives, yielding crude glycal boronic acids[39]. Subsequently, we directly converted them into corresponding borates with various known boronic acid protectecting groups. Following systematic screening and evaluation, we identified C1-glycal N-methyliminodiacetic acid boronates [RB(MIDA)], 1,1,2,2-tetraethylethylene glycol boronates [RB(Epin)], and pinanediol boronates [RB(Pai)], as feasible for preparation on a gram scale, with subsequent purification by silica gel chromatography yielding good yields (see Supplementary Note 2.2 for experimental details). They

exhibit stability against air and moisture and can be stored in the freezer at 4 °C for at least 6 months.

After obtained a series of stable C1-glycal boronates, our subsequent objective was to identify optimal catalytic conditions for achieving a general and robust palladium-catalyzed Suzuki-Miyaura cross-coupling, facilitating the highly modular synthesis of C-1 glycals (Table 1). Encouragingly, treatment of D-glucal pinanediol boronate 12b and methyl 4-bromobenzoate 14a with 10.0 mol% Pd(dppf)$_2$Cl$_2$ and 3.0 equiv K$_3$PO$_4$ in DMF (0.05 M) at 33 °C under N$_2$ for 60 h resulted in a 94% isolated yield of the desired product 15a (entry 1). Using THF, 1,4-dioxane, and toluene as alternative solvents reduced isolated yields to 83%, 79%, and trace amounts, respectively (entry 2). Common phosphine ligand like SPhos, XPhos, and RuPhos yielded similar yields (entries 3–5). Surprisingly, triphenylphosphine also performed excellently, yielding the product almost quantitatively (entries 6 and 7). Reducing the catalyst amount of Pd(dppf)$_2$Cl$_2$ to 5.0 mol% provided a pleasing 95% isolated yield, while 2.5 mol% yielded a 84% isolated yield (entries 8 and 9). Shortening the reaction time to 48 h maintained consistent results (entry 10). Notably, the reaction proceeded under air, albeit with a reduced isolated yield of 57% (entry 11). 83% of Compound 15a was obtained when crude D-glucal-Bpin 12a was used under standard conditions (entry 12). Similar results were obtained using D-glucal 1,1,2,2-tetraethyl dioxaborolane 12c (entry 13), while D-glucal N-methyliminodiacetic acid boronates required appropriate adjustments to the solvent, resulting in reduced yields, approximately 67% (entry 14). This is consistent with the inherently lower reactivity of D-glucal N-methyliminodiacetic acid boronates 12d. To our delight, the robust Pd-catalyzed cross-coupling reaction of efficient C1-glycal boronates processed successfully conducted in a 1:1 mixture of DMF and H$_2$O, yielding 15a at 62% (entry 15). Moreover, increasing the palladium catalyst loading improved the yield to 87% (entry 16). This approach shows promise for late-stage glycodiversification under bioconjugation conditions. As mentioned above, stable C1-glycal boronates 12b and 12c not only enhance stability and purification but also preserve subsequent C-C cross-coupling activity. This is attributed to the strategic spatial protection provided by the four ethyl groups for BEpin and similar stereohindrance for Bpai, which dynamically shield the empty orbital of the boron atom. D-Glucose N-methyliminodiacetic acid boronates 12d, featuring a trivalent ligand N-methyliminodiacetic acid, exhibit enhanced stability due to the strong coordination between the N atom and the B atom. However, this strong coordination significantly reduces their reactivity. Glycal pinacol boronic esters 12a, devoid of bulky protecting groups, exhibit good transmetallation capability due to their unsaturated enol-ether structure's rich electronic properties. However, this structural feature also renders them prone to poor stability and purification properties.

### Reaction scope
With the optimal reaction conditions in hand, the broad scope of various substituted electrophiles in the palladium-catalyzed glycal-based Suzuki-Miyaura cross-coupling was explored. As illustrated in Fig. 2, D-glucal pinanediol boronate 12b demonstrated reactivity with a variety of (hetero)aryl, alkenyl, and alkyl electrophiles, yielding C1-aryl glycals, C1-vinyl glycals, C1- alkynyl glycals, and C1-alkyl glycals in moderate to excellent yields under mild conditions (week base, mild temperature, simple operation). Remarkably, both 4-iodobiphenyl and 4-biphenylyl trifluoromethanesulfonate were well-tolerated, yielding consistent results comparable to 4-bromobiphenyl under optimized reaction conditions (88-98%). Various aryl electrophiles with electron-withdrawing (14a, 14c, 14e-14g) or electron-donating (14b, 14d, 14 h, 14i) substituents on the aryl ring reacted smoothly afforded desired products (15a–15i) in yields ranging from 41 to 98%. Free-protected hydroxyl groups, including phenolic and alcoholic hydroxyl groups, were also well-tolerated. Intriguingly, hinder substrates with electron-donating and electron-withdrawing groups at ortho-positions,

**Table 1 | Optimization of the reaction conditions[a]**

| Entry | Variation from standard conditions | Yield (%) |
|---|---|---|
| 1 | none | 94 |
| 2 | THF, 1,4-dioxane, or toluene instead of DMF | 83, 79, trace |
| 3 | PdCl$_2$/SPhos instead of Pd(dppf)Cl$_2$ | 98 |
| 4 | PdCl$_2$/XPhos instead of Pd(dppf)Cl$_2$ | 95 |
| 5 | PdCl$_2$/RuPhos instead of Pd(dppf)Cl$_2$ | 93 |
| 6 | PdCl$_2$/PPh$_3$ instead of Pd(dppf)Cl$_2$ | 98 |
| 7 | Pd(PPh$_3$)$_2$Cl$_2$ instead of Pd(dppf)Cl$_2$ | 97 |
| 8 | 5 mol% Pd(PPh$_3$)$_2$Cl$_2$ instead of Pd(dppf)Cl$_2$ | 95 |
| 9 | 2 mol% Pd(PPh$_3$)$_2$Cl$_2$ instead of Pd(dppf)Cl$_2$ | 84 |
| 10[b] | 5 mol% Pd(PPh$_3$)$_2$Cl$_2$ instead of Pd(dppf)Cl$_2$ | 96 |
| 11[b,c] | 5 mol% Pd(PPh$_3$)$_2$Cl$_2$ instead of Pd(dppf)Cl$_2$ | 57 |
| 12[b,d] | 5 mol% Pd(PPh$_3$)$_2$Cl$_2$ instead of Pd(dppf)Cl$_2$ | 83 |
| 13[b,e] | 5 mol% Pd(PPh$_3$)$_2$Cl$_2$ instead of Pd(dppf)Cl$_2$ | 92 |
| 14[b,f,g] | 5 mol% Pd(PPh$_3$)$_2$Cl$_2$ instead of Pd(dppf)Cl$_2$ | 67 |
| 15[b,h] | 10 mol% Pd(PPh$_3$)$_2$Cl$_2$ instead of Pd(dppf)Cl$_2$ | 62 |
| 16[b,h] | 50 mol% Pd(PPh$_3$)$_2$Cl$_2$ instead of Pd(dppf)Cl$_2$ | 87 |

[a]Standard reaction conditions: D-glucal pinanediol boronate **12b** (0.12 mmol), **14a** (0.1 mmol), Pd(dppf)$_2$Cl$_2$ (10.0 mol %), K$_3$PO$_4$ (3.0 equiv), DMF (2.0 mL), 33 °C, 60 h, N$_2$, isolated yields. [b]48 h. [c]Under air. [d]Crude D-glucal-Bpin **12a** were used. [e]D-Glucal 1,1,2,2-tetraethylethylene glycol boronates **12c** were used. [f]D-Glucal N-methyliminodiacetic acid boronates **12d** were used. [g]1,4-dioxane/H$_2$O (2.0 mL,1:1). [h]DMF/H$_2$O (2.0 mL,1:1). [B]: Borate esters. TIPS: Triisopropylsilyl groups.

typically challenging in *C*-glycosylations, performed well with our method (**14j–14 m**). Additionally, medicinally relevant heteroarenes such as pyridine (**14n, 14o**) and quinoline (**14p, 14q**) smoothly yielded desired products. Sp²-Hybridized electrophiles extended to alkenyl bromides, furnishing the targeted products (**15r** and **15 s**) in excellent yields with stereospecific Z/E configuration. Notably, palladium-catalyzed C-C cross-couplings of D-glucal pinanediol boronate **12b** with arylethynyl bromide (**14t** and **14 u**) and alkylethynyl bromide (**14 v**) were successfully accomplished under standard conditions, affording desired products in 43–78% isolated yields. Pleasingly, 4-cyanobenzyl bromide and 4-methoxybenzyl bromide are also demonstrated compatibility, delivering **15w** (69%) and **15x** (92%), respectively. Obviously, this palladium-catalyzed Suzuki–Miyaura coupling method efficiently constructs C*sp²*-C*sp*, C*sp²*-C*sp²*, and C*sp²*-C*sp³* bonds simultaneously, even under mild conditions.

Aligned with our focus on the precise preparation of glycopeptides and the modification of complex peptides[53–56], we intend to leverage the robust palladium-catalyzed Suzuki–Miyaura coupling method for efficient synthesis of glycal amino acids or glycal peptides (Fig.3), an area with limited exploration and established synthesis methods[32]. Under standard reaction conditions, both fully protected and partially protected phenylalanine derivatives, including Boc-L-Phe(4-Br)-OMe (**16a**), Boc-L-Phe(4-Br)-OH (**16b**), and NH$_2$-L-Phe(4-Br)-

OMe (**16c**), yielded the corresponding products in 94%, 53%, and 58% yields, respectively. Notably, the versatility of this cross-coupling method in glycopeptide synthesis was convincingly demonstrated by the successful coupling of Phe(4-Br)-containing dipeptides and tripeptides, such as Phe-Phe (**16d**), Phe-Lys (**16e**), Phe-Met (**16 f**), Phe-Thr (**16 g**), Phe-Ser (**16 h**), Phe-Asn (**16i**), Phe-Tyr (**16j**), Phe-Trp (**16k**), and Phe-Leu-Phe (**16 l**), resulting in isolated yields ranging from 43-91%. In Suzuki–Miyaura reactions, substrates with potential containing coordinating groups may impact the cross-coupling reaction, potentially resulting in catalyst poisoning and deboronation. However, we observed that glycal amino acids and peptides products containing free protected groups, such as COOH (**17b**), NH$_2$ (**17c**), OH (**17 g** and **17 h**), and CONH$_2$ (**17i**), are effectively tolerated, affirming the stability and high chemoselectivity of the efficient glycal boronates developed herein under cross-coupling conditions.

To broaden the scope and versatility of our method, we explored the late-stage glycodiversification of various commercially available biologically active molecules and pharmaceuticals using **12b** as the glycal donor under the standard reaction conditions (Fig. 4). Initially, we applied this efficient Pd-catalyzed cross-coupling protocol to core fragments of drugs for type 2 diabetes, such as dapagliflozin, empagliflozin, and ipragliflozin, to prepare precursors of gliflozin drugs (**17m-17o**) in good yields. Moreover, various complex aryl halides

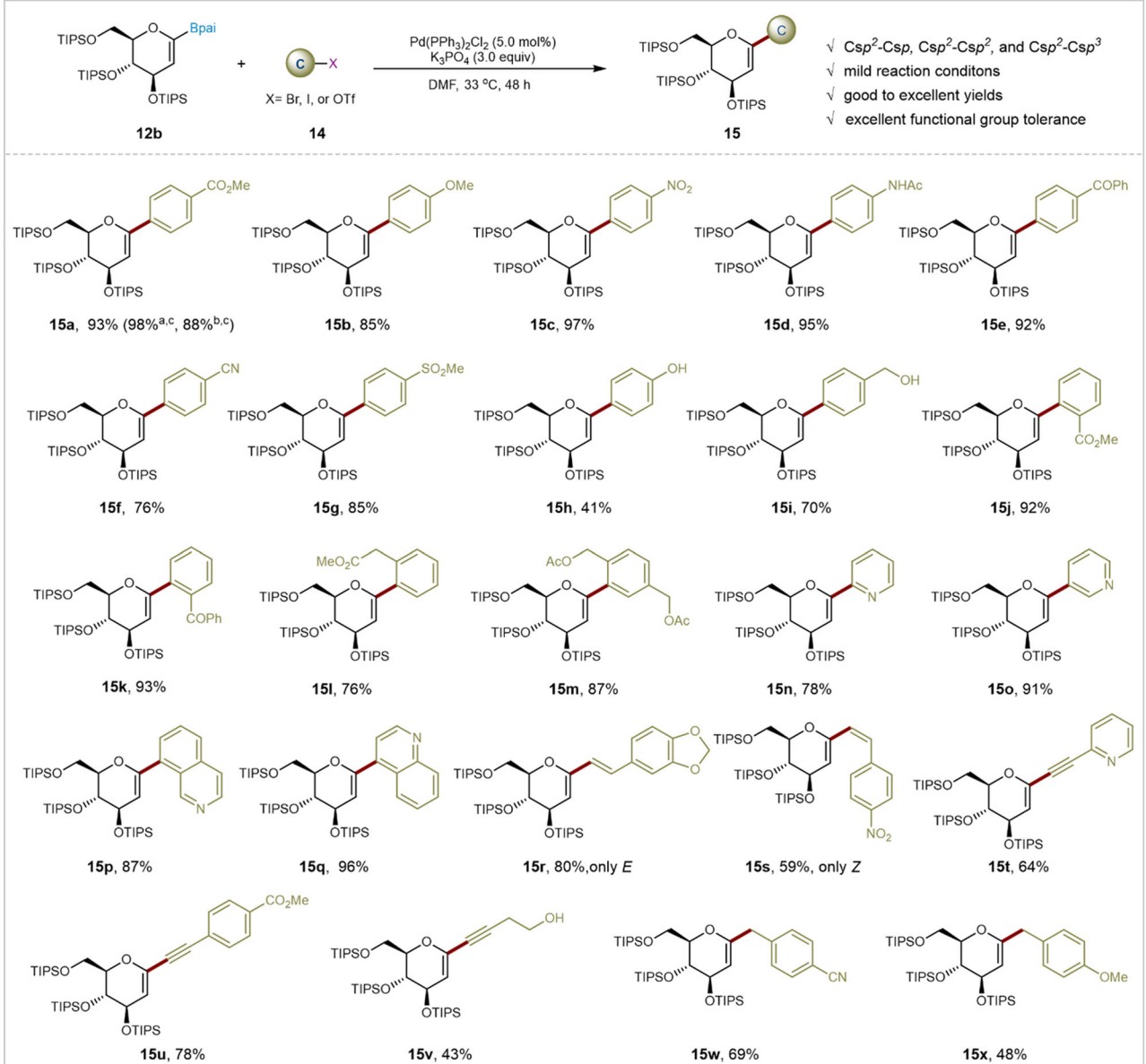

**Fig. 2 | Substrate scope of electrophilic reagents.** General reaction conditions: **12b** (0.12 mmol), **14** (0.10 mmol), Pd(PPh₃)₂Cl₂ (5.0 mol%), K₃PO₄ (3.0 equiv), DMF (2.0 mL), 33 °C, 48 h, N₂, isolated yields. ᵃMethyl 4-iodobenzoate was used. ᵇMethyl 4-(trifluoromethylsulfonyloxy)benzoate was used. ᶜNMR yields. TIPS: Triisopropylsilyl groups. [X]: Halogens or pseudohalogens. C: Carbon electrophiles. Bpai: Pinanediol–boronic acid esters.

derived from commercially available pharmaceuticals and biologically active molecules readily underwent coupling with glucal boronates **12b**. This process facilitated the preparation of glycal-modified drugs, such as the antimicrobial agent sulfadimethoxine (**17p** and **17q**), anti-inflammatory drugs adapalene (**17r**), naproxen (**17t**), and indomethacin (**17w**), the antifungal medication butenafine (**17 u**), the mitogen-activated protein kinase kinase (MEK) inhibitors trametinib (**17 v**), and the hormones dehydroepiandrosterone (**17 s**) and β-estradiol (**17x**) through late-stage functionalization.

The exploration of the substrate scope of metalated glycals, including glycal boronates, has been surprisingly limited. Previous research primarily focused on glucal and galactal substrates, mainly due to the complexity and challenges associated with synthesizing other glycal compounds arising from variations in sugar structures. Recently, Lei's and Yu's groups reported an elegant C-H glycosylation of native carboxylic acids with glycal pinacol esters, utilizing seven different types of glycal pinacol esters in their study[21]. The authors

highlighted an important yet unresolved issue: despite extensive efforts to generate pure glycal using flash column chromatography and other methods, these attempts have consistently yielded fruitless results. This issue may be exacerbated when syntheses need to be scaled. Encouraged by the aforementioned results and these findings, we evaluated various glycal boronates. As depicted in Fig. 5, a wide range of glycal pinanediol and 1,1,2,2-tetraethylethylene glycol boronates, readily synthesized and purified by standard silica gel column chromatography, efficiently underwent coupling reactions, affording the desired products in good to excellent yields. Bridged-silyl protected D-glucal boronate **12e** and TBS-protected D-glucal boronate **12 f** reacted smoothly with both activated and deactivated as well as hindered substrates (**7a-7e**). A comprehensive large-scale test was conducted, and under our standard reaction conditions, we were pleased to observe that the yields of **7b** and **7d** were consistently maintained at approximately 90%. Subsequently, a variety of glycal 1,1,2,2-tetraethylethylene glycol (BEpin) and glycal

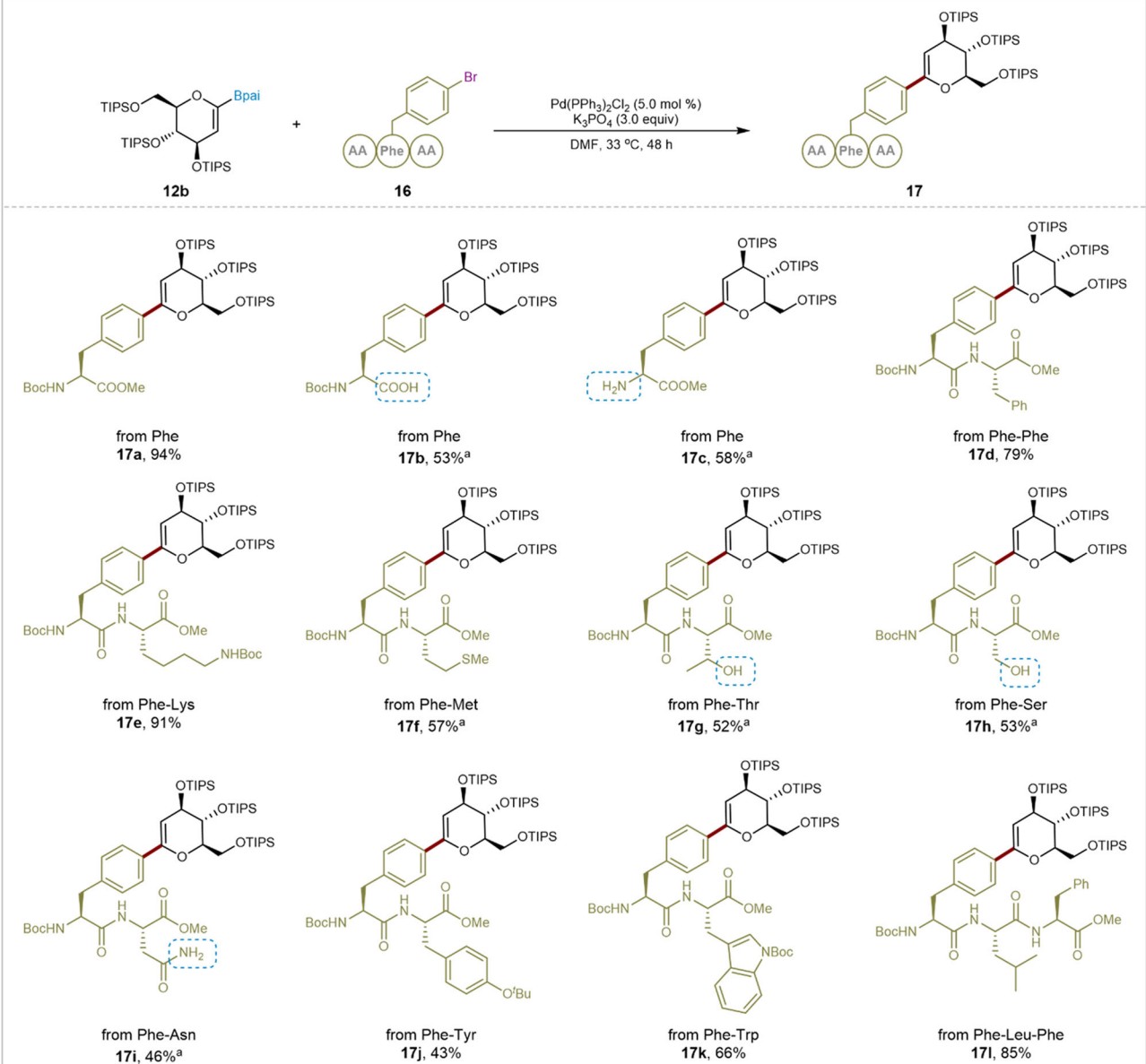

**Fig. 3 | Substrate scope of oligopeptide electrophilic reagents.** General reaction conditions: **12b** (0.12 mmol), **16** (0.10 mmol), Pd(PPh₃)₂Cl₂ (5.0 mol%), K₃PO₄ (3.0 equiv), DMF (2.0 mL), 33 °C, 48 h, N₂, isolated yields. ᵃ50.0 mol% Pd(PPh₃)₂Cl₂ was used. reaction conditions, showcasing the favorable balance of reactivity and stability exhibite by the glycal boronates and their potential for broader impacts. TIPS: Triisopropylsilyl groups. AA: Amino acids. Bpai: Pinanediol–boronic acid esters.

pinanediol (Bpai) boronates deveried from common pyranoses, such as D-galactose, L-rhamnose, 6-deoxy-D-glucose, and D-arabinose, were readily transformed to the corresponding *C*1-aryl glycals (**7f-7j**) in excellent yields. To our delight, the glycal boronates were synthesized from the corresponding disaccharides and delivered the coupled products in excellent yields, showcasing the feasibility of directly installing polysaccharides. It is noteworthy that glycal 1,1,2,2-tetraethylethylene glycol (BEpin) borate has been demonstrated to be more robust overall compared to glycine pinanediol (Bpai) borate.

To underscore the exceptional reactivity, robustness, and compatibility of these developed glycal boronates, we seamlessly conducted one-pot reactions involving Ir-catalyzed glycal borylation followed by Pd-catalyzed Suzuki–Miyaura coupling. This streamlined approach offers a direct pathway for synthesizing a range of valuable *C*1-aryl glycals, demonstrating the method's adaptability to diverse scenarios (Fig. 6).

## Application

DNA-encoded libraries (DELs) present a promising platform for hit identification across academia and pharmaceutical industry[57]. Constructing structurally diverse and pharmaceutically relevant DELs necessitates the development of more DNA-compatible reactions[58]. Given their structural diversity and biological activity, incorporating sugar moieties onto DNA tags to create sugar-containing DELs holds significant promise for the R&D of carbohydrate-based drug discovery and chemical biology research[3]. However, due to the complexity of sugar molecules and the scarcity of DNA-compatible glycosylations, chemical reactions linking sugars with DNA tags are rare[59–62]. Encouraged by the successful results and considering the excellent reactivity and stability of the efficient *C*1-glycal boronates here developed, we integrated DEL technology with our highly efficient Pd-catalyzed Suzuki–Miyaura coupling methods to synthesize carbohydrate-DNA conjugates (Fig. 7). With slight modifications of reaction conditions using water-soluble palladium precatalyst sSPhos-Pd-G2[63], DMF:

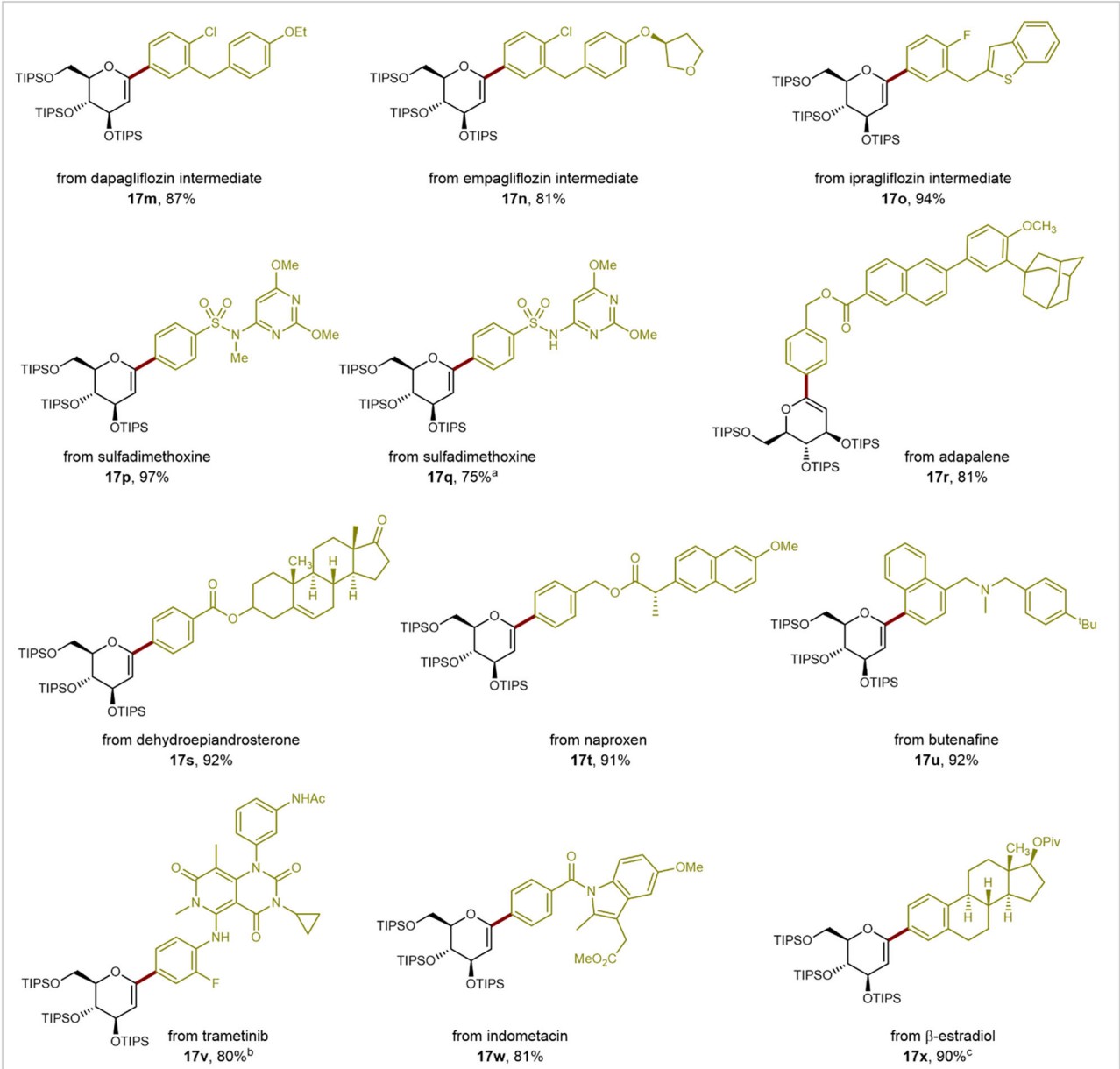

**Fig. 4 | Scope with core structures of bioactive molecules and pharmaceuticals.** General reaction conditions: **12b** (0.12 mmol), bioactive molecules or pharmaceuticals (0.10 mmol), Pd(PPh₃)₂Cl₂ (5.0 mol%), K₃PO₄ (300.0 mol%), DMF (2.0 mL), 33 °C, 48 h, N₂, isolated yields. [a]10.0 mol% Pd(PPh₃)₂Cl₂ was used. [b]Aryl iodine reagent were used. [c]Trifluoromethanesulfonates were used. TIPS: Triisopropylsilyl groups.

Dioxane: EtOH (1:1:1) as organic co-solvent, and CsOH as the base, DNA headpieces bearing with various aryl iodides and bromides were successfully transformed into glycal-DNA conjugates using bridged-silyl protected D-glucal boronate (**20a-20h**). Remarkably, this reaction was effective across various highly coordinating heterocyclic substrates, including pyridines (**19b**) and (**19 h**), furan (**19c**), thiazole (**19e**), Imidazo[1,2-a]pyridine (**19d**), and (**19 f**), and pyrazole (**19 g**). Surprisingly, DNA headpieces containing aryl chlorides seamlessly underwent coupling with comparable conversion rates (**20a** and **20i**), suggesting the adjustable reactivity of glycal boron reagents under varying conditions. A series of various glycal-modified DNA conjugates, including D-arabinose (**20j-20m**), L-rhamnose (**20o- 20q**), D-galactose (**20r** and **20 s**) were obtained in good yields. Again, various DNA headpieces bearing aryl chlorides also smoothly coupled with various glycal boronates to obtain final coupling products (**20a, 20i, 20 m, 20q**, and **20 s**). To our knowledge, this method marks the important use of

glycal-based Suzuki–Miyaura coupling with aryl electrophiles and for synthesizing glycal-DNA conjugates, highlighting its potential for significant applications in drug discovery and chemical biology research.

The synthetic value of such cross-coupling is further highlighted by its application in the total synthesis of carbohydrate-base drugs and the glycodiversification of various drugs (Fig. 8). Initially, the glycal-based Suzuki–Miyaura coupling was employed in the synthesis of dapagliflozin (**7c-2**), an approved inhibitor for treating type 2 diabetes. Following the standard procedure, C1-aryl-D-glucal (**7c-1**) was synthesized with a yield of 92%. Subsequently, dapagliflozin (**7c-2**) was obtained in 78% isolated yield with exclusive β-anomeric stereoselectivity through hydroboration-oxidation and deprotection of the silyl protecting group (Fig. 8a). The configuration of the anomeric carbon in C-aryl glycosides was determined by analyzing the $^3J_{(HH)}$ coupling constants of the H1 proton signal, typically found in the range of 3.81-5.25 ppm (CDCl₃). For 1,2-trans isomers, these

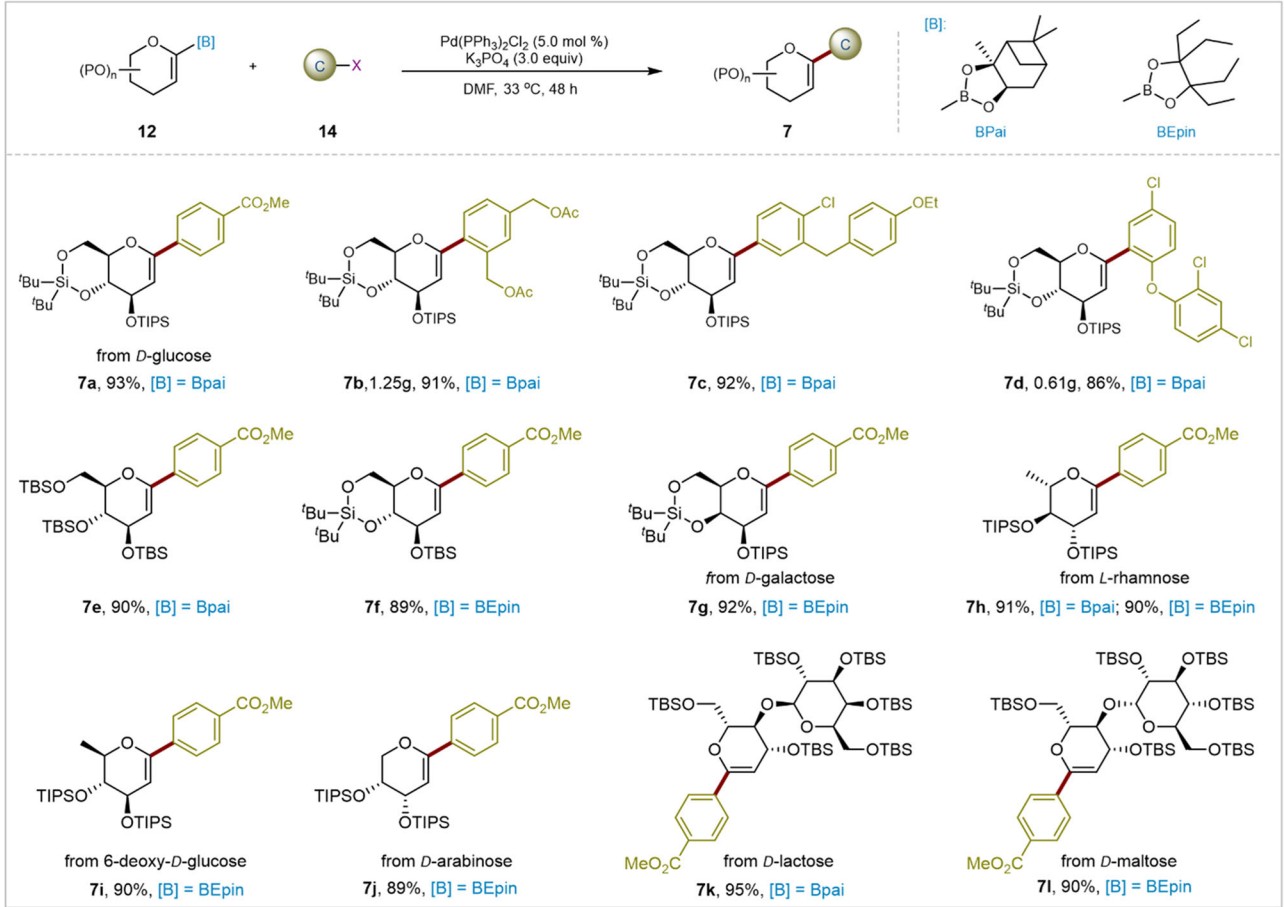

**Fig. 5 | Substrate scope of glycal boronates.** Reaction conditions: **12** (0.12 mmol), **14** (0.10 mmol), Pd(PPh₃)₂Cl₂ (5.0 mol%), K₃PO₄ (3.0 equiv), DMF (2.0 mL), 33 °C, 48 h, N₂, isolated yields. [B]: Borate esters. P: Protecting groups. [X]: Halogens or pseudohalogens. C: Carbon electrophiles. Bpai: Pinanediol–boronic acid esters. BEpin: 1,1,2,2-Tetraethylethylene glycol–boronic acid esters. TIPS: Triisopropylsilyl groups. TBS: *tert*-Butyl dimethylsilyl groups.

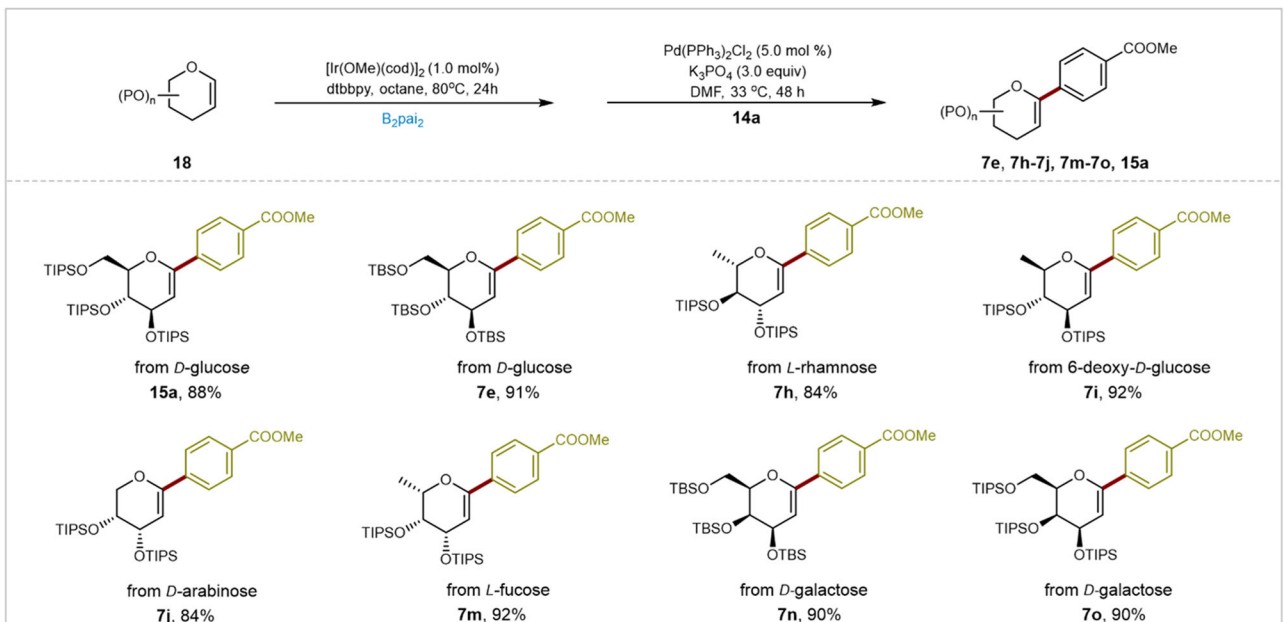

**Fig. 6 | The one-pot reaction involves the Ir-catalyzed borylation of glycal followed by the Pd-catalyzed Suzuki–Miyaura coupling.** Reaction conditions: **18** (0.18 mmol, 1.8 equiv), B₂Pai₂ (0.12 mmol, 1.2 equiv), [Ir(OMe)(cod)]₂ (2.0 mol%), dtbbpy (4.0 mol%), *n*-octane (1.0 ml), 80 °C, 24 h, then **14** (0.10 mmol, 1.0 equiv), Pd(PPh₃)₂Cl₂ (5.0 mol%), K₃PO₄ (3.0 equiv), DMF (2.0 mL), 33 °C, 48 h, N₂, isolated yields. P: Protecting groups. TIPS: Triisopropylsilyl groups. TBS: *tert*-Butyl dimethylsilyl groups.

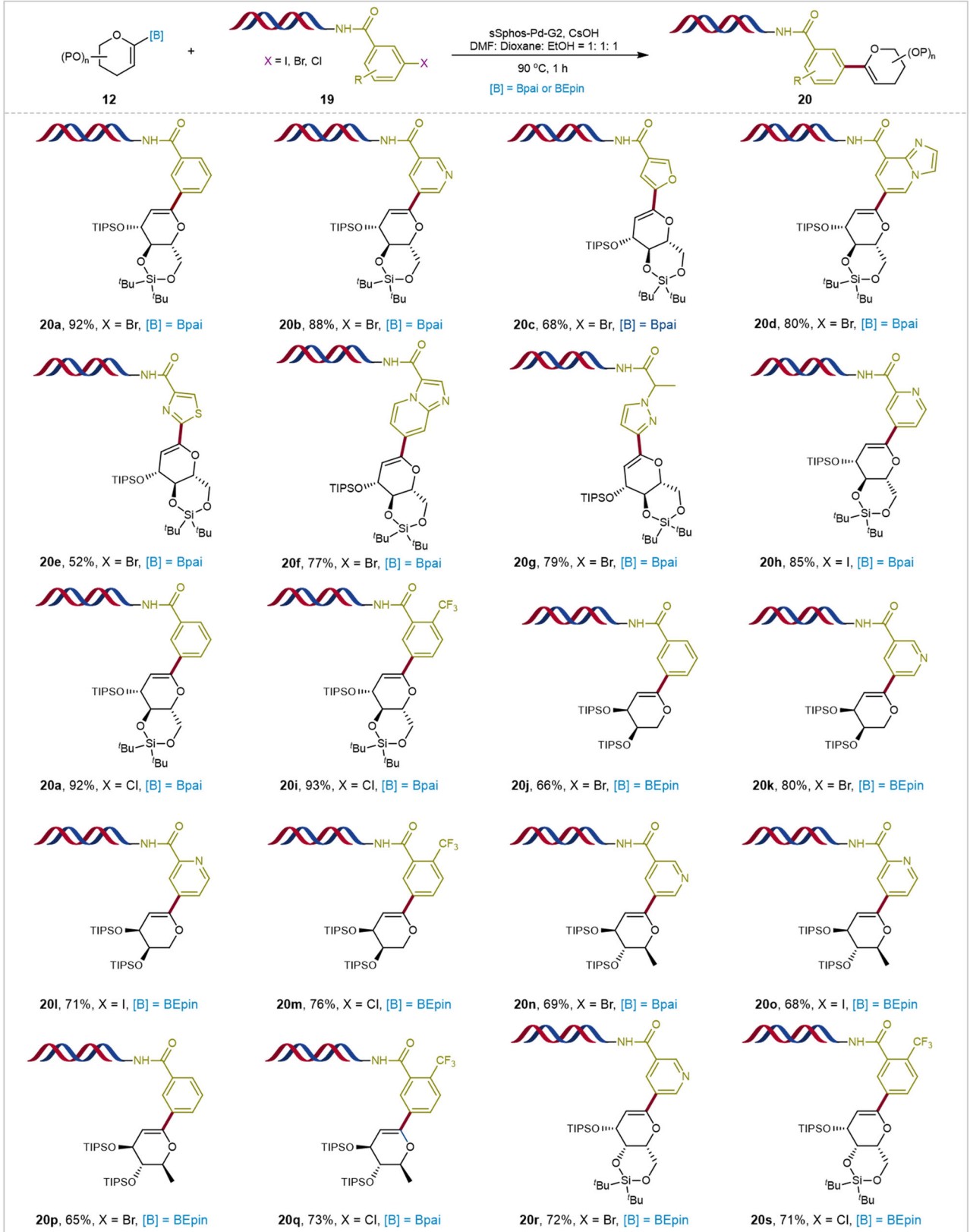

**Fig. 7 | On-DNA glycal-based Suzuki−Miyaura coupling of (hetero)aryl halides.**
Reaction conditions: DNA headpieces **19** (1.00 equiv), CsOH (100 equiv), glycal
boronates **12** (250 equiv, 0.15 M in DMF: Dioxane: EtOH = 1: 1: 1), and sSphos-Pd-G2
(2.00 equiv), 90 °C, 1 h, the conversion was determined by LC-MS. [B]: Borate esters.

P: Protecting groups. [X]: Halogens or pseudohalogens. Bpai: Pinanediol−boronic
acid esters. BEpin: 1,1,2,2-Tetraethylethylene glycol−boronic acid esters. TIPS:
Triisopropylsilyl groups.

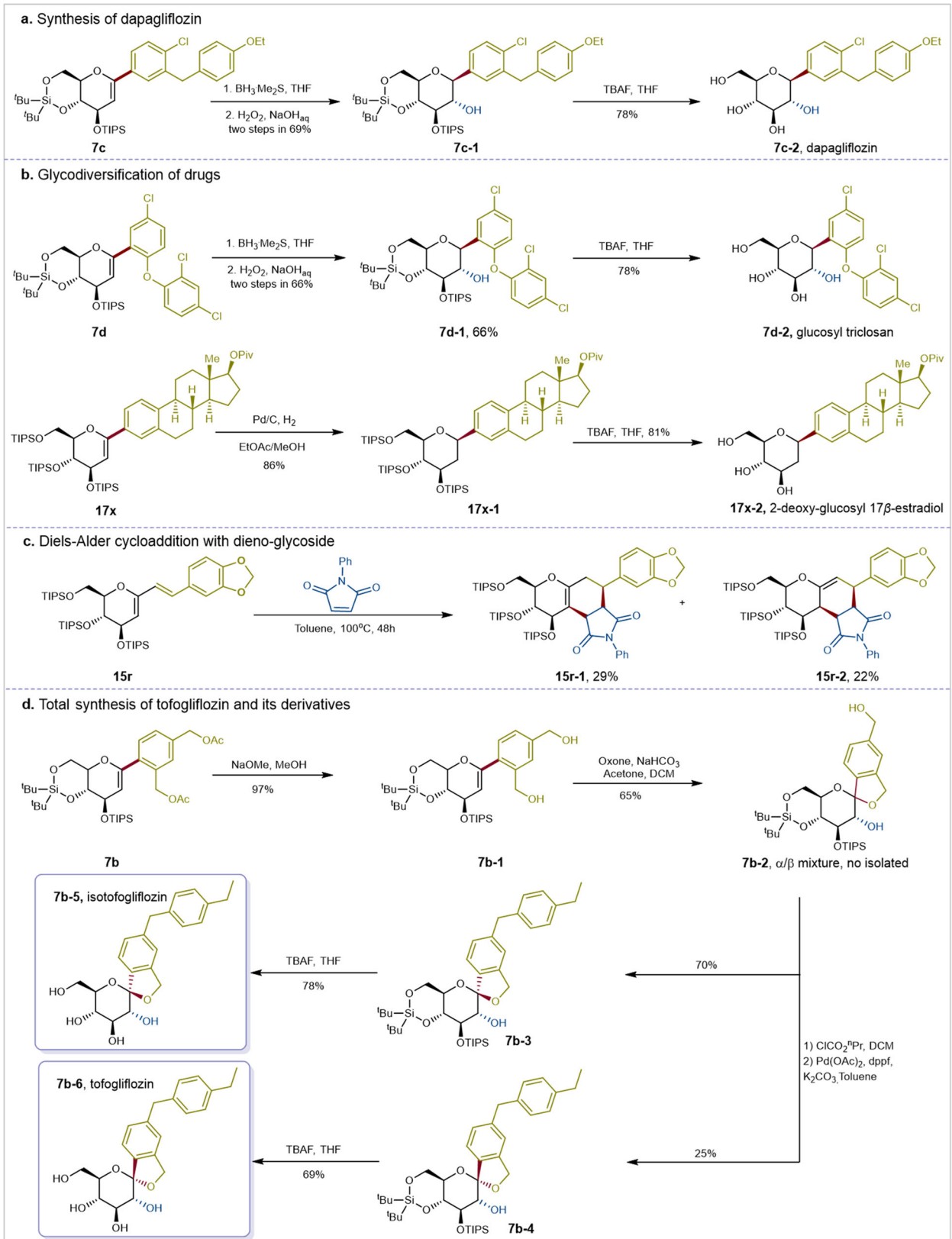

**Fig. 8 | Late-stage glycodiversifications and total synthesis of pharmaceuticals and biologically active molecules. a** Synthesis of dapagliflozin. **b** Glycodiversification of triclosan and 17β-estradiol. **c** Diels-Alder cycloaddition with dieno-glycoside. **d** Total synthesis of tofogliflozin and its derivatives. See Supplementary Note 2.6 for experimental details. TIPS: Triisopropylsilyl groups.

coupling constants fall in the range of 8.80-9.40 Hz, while for 1,2-*cis* isomers, they typically range from 4.20-6.20 Hz. For instance, the $^3J_{(HH)}$ coupling constant of the H1 proton signal of compound **7c-1** was located at 4.19 ppm (500 MHz, CDCl$_3$) with a value of 9.4 Hz, indicating a 1,2-trans configuration, thus β-configuration. Late-stage glycodiversifications of commercially available pharmaceuticals and biologically active molecules offers a approach to develop efficient bioactive molecules. For instance, D-glucal triclosan (**7d**) subjected to hydroboration-oxidation and subsequent deprotection of the silyl protecting group, yielding D-glucosyl triclosan (**7d-2**) in 78% isolated yield with exclusive β-anomeric stereoselectivity (Fig. 8b, top). Palladium-catalyzed stereoselective hydrogenation of D-glucal *β*-estradiol (**17x**) efficiently afforded the intermediate (**17x-1**), which underwent deprotection of the silyl protecting group to afford the final 2-dexoy-D-glucosyl *β*-estradiol derivative (**17X-2**) in 81% with exclusive β-anomeric stereoselectivity (Fig. 8b, bottom). Pleasingly, the Diels-Alder Reaction of *C1*-vinyl glucal **15r** and *N*-Phenylmaleimide also proceeded smoothly, yielding two tricyclic *C*-glycosides, **15r-1** and **15r-2**, with endo stereoselectivity (Fig. 8c). Compound **15r-1** is the product of the double bond isomerization of **15r-2**. Hydrolysis of the two OAc groups of *C1*-aryl D-glucal (**7b**) resulted in nearly quantitative formation of the intermediate (**7b-1**). Subsequent conversion involved oxidation of the double bond with in situ prepared dimethyldioxirane (DMDO) under mild conditions, yielding a mixture of *α* and *β*-anomers (**7b-2**) without deliberate separation. The hydroxyl group in benzyl position of the intermediate was then activated and transfromed to an ethylphenyl group via Pd-catalyzed Suzuki–Miyaura coupling with 4-ethylphenylboronic acid, resulting in the isolable silyl-protected *α*-tofogliflozin (**7b-3**) in 70% yield and *β*-tofogliflozin (**7b-4**) in 25% yield. After deprotection of the silyl protecting group, isotofogliflozin (**7b-5**) was obtained in 78% yield and tofogliflozin (**7b-6**) in 69% yield (Fig. 8d).

In summary, our study marks an important for advancement in the development of glycal boronates, employing 1,1,2,2-tetra-ethylethylene glycol and pinanediol as protective groups. These boronates boaste good reaction reactivity, stability, and ease to handling. A diverse range of structurally varied glycal boronates, including common monosaccharides and disaccharides, can be synthesized in gram-scale yields and purified by standard silica gel chromatography with ease. Leveraging their balanced reactivity and stability, we successfully established a robust palladium-catalyzed Suzuki-Miyaura cross-coupling with diverse (hetero)aryl, alkenyl, and alkyl electrophiles, facilitating the efficient assembly of C$sp^2$-C$sp$, C$sp^2$-C$sp^2$, and C$sp^2$-C$sp^3$ bonds in good to excellent yields under mild reaction conditions. Demonstrating its broad applicability, we showcased over 80 examples, including oligosaccharides, oligopeptides, pharmaceuticals, and biologically active molecules. Enabled by this methodology, numerous glycal and glycosyl modifed pharmaceuticals, along with their analogues, were smoothly obtained through direct late-stage glycosylation or total synthesis. Notably, our approach represents the efficient instance of glycal-based Suzuki–Miyaura coupling for synthesizing glycal-DNA conjugates, thereby enriching the library of DNA-compatible glycosylations. We anticipate that palladium-catalyzed Suzuki-Miyaura cross-coupling of stable glycal boronates will emerge as a general, simple, and robust tool for the flexible, scalable synthesis of *C*-glycosyl complex pharmaceuticals and other medically relevant compounds. Additionally, this contributes to the broader application of organoboron compounds in biomolecules and advances organoboron chemistry research.

## Methods

### General procedure for palladium-catalyzed Suzuki-Miyaura cross-coupling

Glycal boronates (1.20 equiv), electrophilic reagents (1.00 equiv), Pd(PPh$_3$)$_2$Cl$_2$ (5.0 mol%), K$_3$PO$_4$ (3.00 equiv) were added to a one-dram vial with a screw-top septum, and the vial was then evacuated and refilled with N$_2$ (3×). Anhydrous DMF (2.00 mL) were added, and the reaction mixture was stirred at 33 °C for 48 h, cooled to rt, and concentrated. The crude material was purified by column chromatography on SiO$_2$.

### General procedure for one pot reaction of Ir-catalyzed borylation of glycal and palladium-catalyzed Suzuki–Miyaura coupling

Glycals (1.80 equiv), B$_2$Pai$_2$ or B$_2$Epin$_2$ (1.20 equiv), [IrOMe(cod)]$_2$ (2.4 mol%), dtbbpy (4.8 mol%) were added to a one-dram vial with a screw-top septum, and the vial was then evacuated and refilled with N$_2$ (3×). Anhydrous octane (1 mL) was added, and the reaction mixture was stirred at 80°C for 24 h. The reaction mixture cooled to rt and concentrated under vaccum, then electrophilic reagent (1.00 equiv), Pd(PPh$_3$)$_2$Cl$_2$ (5.0 mol%), K$_3$PO$_4$ (3.00 equiv) were added, evacuated and refilled with N$_2$ (3×). Anhydrous DMF (2.00 mL) were added, and the reaction mixture was stirred at 33 °C for 48 h, cooled to rt, and concentrated. The crude material was purified by column chromatography on SiO$_2$.

### General Procedure for On-DNA glycal-based Suzuki–Miyaura coupling of (hetero)aryl halides

To each well of a 250 µL 96-well PCR microplate was added DNA 1 (1.00 equiv, 10.0 nmol, 10.0 µL, 2.00 mM in H$_2$O), CsOH (100 equiv, 1.00 µmol, 2.00 µL, 0.50 M in H$_2$O), glycal boronates (250 equiv, 2.50 µmol, 16.70 µL, 0.15 M in DMF: Dioxane: EtOH = 1: 1: 1), and sSphos-Pd-G2 (2.00 equiv, 20.00 nmol, 1.00 µL, 20.00 mM in DMF) sequentially. The solution was mixed with vortex for 30 s, then reacted at 90 °C for 1 h. When time's up, adding DDTC (100 equiv, 1.00 µmol, 2.00 µL, 0.50 M in H$_2$O) to each well, the solution was mixed with vortex for 30 s, then reacted at 30 °C for 10 min. When time's up, centrifuging under 4 °C with 4000 rpm for 10 min. Remove precipitates and take supernatant for further ethanol precipitation, add 10% (v/v) 5 M NaCl solution and 3 times the volume of absolute ethanol to supernatant, cooled under −78 °C for 2 h. Centrifuge under 4 °C with 4000 rpm for 30 min. The precipitated material was isolated as a pellet by centrifugation and subsequent removal of the supernatant. 75% aq. ethanol was then added to the pellet and the mixture was centrifuged again. The supernatant again was discarded and the DNA pellet was dried under vacuum. The DNA pellet was redissolved in H$_2$O as 0.5 mM. Then take 1-2 nmol DNA for LC-MS detection.

## Data availability

The authors declare that the data supporting the findings of this study, including experimental details and compound characterization, are available within the article and its Supplementary Information file. All data are available from the corresponding author upon request.

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

## Acknowledgements

We are grateful for financial support from the National Key R&D Program of China (Grant No. 2023YFA1508800, F. Z.), National Science Foundation (Grant No. 22301178, F. Z.), Shanghai Pilot Program for Basic Research - Shanghai Jiao Tong University (Grant No. 21TQ1400210, F. Z.), Fundamental Research Funds for the Central Universities (Grant No. 22×010201631, F. Z.), the Open Grant from the Pingyuan Laboratory (Grant No. 2023PY-OP-0102, F. Z.), Natural Science Foundation of Shanghai (Grant No. 21ZR1435600, F. Z.), Shanghai Sailing Program (Grant No 21YF1420600, F. Z.). Part of this study was supported by the National Science Foundation (Grant No. 22301180, B. Y.).

## Author contributions

F.Z. conceived and supervised the project. A.C., Y.H., and R.W. performed the experiments and analyzed the data. F.Z., L.Z., A.C., and B.Y. discussed the results and wrote the manuscript with input from all authors. All authors have read and approved the final version of the manuscript.

## Competing interests

The authors declare no competing interests.
