## [Peer Review File · Nature Communications]

Palladium-catalyzed Suzuki-Miyaura cross-couplings of stable glycal boronates for robust synthesis of C-1 glycalsREVIEWER COMMENTS

Reviewer #1 (Remarks to the Author):

The authors of this manuscript have developed a type of stable glycal boronates that can undergo a robust palladium-catalyzed glycal-based Suzuki-Miyaura reaction. This reaction enables the formation of various types of bonds including C(sp²)-C(sp), C(sp²)-C(sp²), and C(sp²)-C(sp³) under mild conditions. The versatility of this reaction has been demonstrated in over 80 examples, including oligosaccharides, oligopeptides, pharmaceuticals, and biologically active molecules. The authors have also successfully created glycal-DNA conjugates for the first time. Additionally, this reaction has been employed in late-stage glycosylation and the total synthesis of glycosyl pharmaceuticals and analogues.

The manuscript is well-written, the chemistry is sound, and the study is of high importance and non-trivial to conduct. The study may be eligible for publication in Nature Communications after addressing some concerns.

1. The authors stated that the C1-glycal boronates were 'feasible for preparation on a gram scale', but only part of the involved boronates was prepared 'on a gram scale' according to the SI.
2. Table 1, the authors aim to demonstrate the properties of C1-glycal boronates regarding stability, transmetallation, and purification. However, the readers might get confused about the order of the information presented in the scheme. For instance, it's not clear whether the right or left side of the Purification row represents the easier or more difficult properties.
3. All tables and figures related to product yields must clearly indicate the method of yield calculation (isolated yield, HPLC yield or determined by LC-MS) as footnotes.
4. It is suggested that the authors provide a brief explanation of how the anomeric configuration of the glycosyl residue was determined after the "glycodiversifications," for example, the compound 7c-1.

Reviewer #2 (Remarks to the Author):

C-glycoside represents a class of carbohydrate that exhibit superior stability and bioavailability than conventional O-glycosides. The synthesis of C-glycosides has attracted considerable attentions in the field of synthetic chemistry and medicinal chemistry. In this manuscript, the author reported an efficient and highly applicable method for the synthesis of C-glycals by palladium-catalyzed Suzuki-Miyaura coupling reaction, and the C-glycals can be converted to diversified C-glycosides by downstream transformations. The newly synthesized stable glycal boronates enable the efficient coupling with a broad range of aryl, alkenyl and alkyl halides. Most importantly, this method showcased high compatibility with bio-active reagents, including oligopeptides and DNA headpieces. Overall, this procedure provides a streamlined strategy for the synthesis of C-glycosyl modified pharmaceuticals and relevant analogues. This reviewer would like to suggest the acceptance of the manuscript for publication in Nat. Commun.

1. Since unprotected hydroxyl group is tolerated in the Suzuki-Miyaura coupling, it's worth trying to remove the protecting groups of glycal boronates and conducting the Suzuki-Miyaura coupling with unprotected glycal boronates. This will further enhance the applicability of the developed method.
2. In Table 1, the yield of the coupling reaction between 12a and 14a should be provided as a standard of comparison.

3. As 2 equivalents of sSphos-Pd-G2 were used in the reaction with DNA headpieces, the authors are encouraged to analyze the residual amount of palladium in the final product.

Reviewer #3 (Remarks to the Author):

Zhu et al. describe the Pd-catalyzed Suzuki–Miyaura couplings of novel glycal boronates which were prepared by Ir-catalyzed glycal C–H borylation using diboron reagents, B2pai2 or B2Epin2. The prepared 1,1,2,2-tetraethylethylene glycole glycal boronates (RBEpin) and pinanediol boronates (RBpai) were quite stable and easy to handle. They applied this method for the synthesis of a wide variety of compounds such as peptides, derivatives of pharmaceuticals, DNA-encoded compounds, and natural products. Although those might become an important portion of work for boron and glycal chemistry, they did not directly compare their method with the previously reported related method (Ref 38) using the most common pinacol boronates. The authors need to show clearly how difficult to handle pinacol esters for glycal chemistry using experimental facts because the prior literature (Ref 38) did not mention anything about the instability of pinacol esters.

For the above reason, this reviewer could not recommend this manuscript for publication in Nature Communications.

This reviewer also found:

1. The missing explanation of the comparison among Bpin, Bpai, BEpin, and BMIDA in the text would be a serious problem although the authors illustrate the difference in stabilities, abilities of transmetalation, and handlings during purification among them in Table 1 without supporting data.

We would like to express our gratitude to the Editor and the Reviewer for dedicating their time and providing valuable comments on our manuscript. Their insights have been instrumental in enhancing the quality of this work. We have incorporated all of the reviewers' comments in our revised manuscript. Please find the revised manuscript and Supporting Information with the highlighted changes.

Point-by-point responses to the referees' comments is as follows:

Reviewer: 1

The authors of this manuscript have developed a type of stable glycal boronates that can undergo a robust palladium-catalyzed glycal-based Suzuki-Miyaura reaction. This reaction enables the formation of various types of bonds including C(sp²)-C(sp), C(sp²)-C(sp²), and C(sp²)-C(sp³) under mild conditions. The versatility of this reaction has been demonstrated in over 80 examples, including oligosaccharides, oligopeptides, pharmaceuticals, and biologically active molecules. The authors have also successfully created glycal-DNA conjugates for the first time. Additionally, this reaction has been employed in late-stage glycosylation and the total synthesis of glycosyl pharmaceuticals and analogues.

The manuscript is well-written, the chemistry is **sound**, and the study is of **high importance** and **non-trivial to conduct**. The study may be **eligible** for publication in Nature Communications after addressing some concerns.

Response: We express our heartfelt gratitude to the Reviewer for their positive evaluation and support for publication.

1. The authors stated that the C1-glycal boronates were 'feasible for preparation on a gram scale', but only part of the involved boronates was prepared 'on a gram scale' according to the SI.

Response: We sincerely appreciate the valuable suggestions provided by the Reviewer. In response to this insightful input, in addition to **12b** (1.10 g), **12g** (0.90 g), **12h-2** (1.40 g), **12i** (0.79 g), **12j** (0.89 g), and **12m** (2.08 g), which exceed or approach gram-scale quantities, we have conducted additional experiments to prepare C1-glycal boronates on a gram scale, including **12c** (1.43 g), **12e** (1.60 g), and **12h-1** (1.75 g). These examples further demonstrate the capability to prepare and isolate stable glycal boron reagents on a large scale. The new results have been updated to the revised Supporting Information.

2. Table 1, the authors aim to demonstrate the properties of C1-glycal boronates regarding stability, transmetallation, and purification. However, the readers might get confused about the order of the information presented in the scheme. For instance, it's not clear whether the right or left side of the Purification row represents the easier or more difficult properties.

Response: We are deeply grateful for the invaluable suggestion provided by the Reviewer. In response to their input, we have implemented revisions to the scheme

outlined in Table 1, which are now reflected in the updated manuscript. Enclosed below is the revised scheme for your reference:

3. All tables and figures related to product yields must clearly indicate the method of yield calculation (isolated yield, HPLC yield or determined by LC-MS) as footnotes.

Response: We sincerely thank the reviewers for bringing up this crucial question. We have ensured that all tables and figures concerning product yields are accompanied by footnotes, clearly indicating the method of yield calculation employed, whether it be isolated yield, HPLC yield, or determined by LC-MS. We appreciate the attention to detail and are grateful for highlighting this aspect, which has been duly addressed in our manuscript.

4. It is suggested that the authors provide a brief explanation of how the anomeric configuration of the glycosyl residue was determined after the "glycodiversifications," for example, the compound **7c-1**.

Response: We are sincerely grateful for the valuable suggestion from the Reviewer. In response to this insightful input, we have included a brief explanation of how the anomeric configuration of the glycosyl residue was determined after the "glycodiversifications" in the revised manuscript. Below is the relevant passage:

*"The configuration of the anomeric carbon in C-aryl glycosides was determined by analyzing the ³J_(HH) coupling constants of the H1 proton signal, typically found in the range of 3.81-5.25 ppm (CDCl₃). For 1,2-trans isomers, these coupling constants fall in the range of 8.80-9.40 Hz, while for 1,2-cis isomers, they typically range from 4.20-6.20 Hz. For instance, the ³J_(HH) coupling constant of the H1 proton signal of compound **7c-1** was located at 4.19 ppm (500 MHz, CDCl₃) with a value of 9.4 Hz, indicating a 1,2-trans configuration, thus β-configuration. "*

Reviewer 2:

C-Glycoside represents a class of carbohydrate that exhibit superior stability and bioavailability than conventional O-glycosides. The synthesis of C-glycosides has attracted considerable attentions in the field of synthetic chemistry and medicinal

chemistry. In this manuscript, the author reported an efficient and highly applicable method for the synthesis of C-glycals by palladium-catalyzed Suzuki-Miyaura coupling reaction, and the C-glycals can be converted to diversified C-glycosides by downstream transformations. The **newly** synthesized **stable** glycal boronates enable the efficient coupling with a broad range of aryl, alkenyl and alkyl halides. Most importantly, this method showcased **high compatibility** with bio-active reagents, including oligopeptides and DNA headpieces. Overall, this procedure provides a streamlined strategy for the synthesis of C-glycosyl modified pharmaceuticals and relevant analogues. This reviewer would like to suggest the acceptance of the manuscript for publication in *Nat. Commun.*

Response: We want to convey our sincere appreciation to the Reviewer for their positive evaluation and unwavering support for publication.

1. Since unprotected hydroxyl group is tolerated in the Suzuki-Miyaura coupling, it's worth trying to remove the protecting groups of glycal boronates and conducting the Suzuki-Miyaura coupling with unprotected glycal boronates. This will further enhance the applicability of the developed method.

Response: We wholeheartedly agree with the Reviewer's perspective. As depicted in Fig. 2, our experimental results (15h, 15i, and 15v) demonstrate that unprotected hydroxyl groups are tolerated in the glycal-based Suzuki-Miyaura coupling.

In fact, during the preparation of this manuscript, we made **extensive efforts** to remove the silicon-based protecting groups from glycal boronates. Despite attempting the commonly used method involving TBAF (tetrabutylammonium fluoride), we were unfortunately unable to obtain unprotected glycal boronates. The decomposition of glycal boronates observed under TBAF treatment may be attributed to the rich electronic properties of their unsaturated enol-ether structure.

Again, we greatly appreciate the Reviewer's valuable suggestion.

2. In Table 1, the yield of the coupling reaction between 12a and 14a should be provided as a standard of comparison.

Response: We deeply appreciate the reviewers for raising this important question. In response to this insightful input, we have employed curde **12a** and **14a** to conduct the reaction under the standard conditions, and indeed, the desired product was obtained in 83% yield. The updated result has been incorporated into Table 1 of the revised manuscript.

3. As 2 equivalents of sSphos-Pd-G2 were used in the reaction with DNA headpieces, the authors are encouraged to analyze the residual amount of palladium in the final product.

Response: We sincerely appreciate the valuable suggestion from the Reviewer. For individual reactions, the residues of palladium catalyst may vary depending on the reaction substrates. To eliminate any potential palladium residue across different

substrate systems, we employ a post-reaction treatment with excess sodium diethyldithiocarbamate (DDTC, 100 equiv). This addition facilitates the capture of palladium in the reaction system. Subsequently, the formed dark brown Pd-DDTC precipitate is removed via centrifugation. The supernatant is then subjected to ethanol precipitation to further purify the system (DDTC remove Pd: Ref. *Org. Lett.* **2023**, 25, 37, 6931–6936; *Bioconjugate Chem.* **2021**, 32, 11, 2290–2294).

Regarding the construction and production of DNA-encoded libraries (DELs), in addition to standard chemical workup procedures, various methods are employed, including multiple rounds of ethanol precipitation, ultrafiltration, HPLC purification, and bead-based purification. These meticulous purification steps minimize the presence of palladium residues in the system. This updated information has been included in the Supporting Information.

Reviewer 3:

Zhu et al. describe the Pd-catalyzed Suzuki–Miyaura couplings of novel glycol boronates which were prepared by Ir-catalyzed glycol C–H borylation using diboron reagents, B₂pai₂ or B₂Epin₂. The prepared 1,1,2,2-tetraethylethylene glycol glycol boronates (RBEpin) and pinanediol boronates (RBpai) were quite stable and easy to handle. They applied this method for the synthesis of a wide variety of compounds such as peptides, derivatives of pharmaceuticals, DNA-encoded compounds, and natural products.

Although those might become an important portion of work for boron and glycol chemistry, they did not directly compare their method with the previously reported related method (Ref 38) using the most common pinacol boronates.

Response: We appreciate the Reviewer for bringing up this question. To address the concern, we conducted Suzuki–Miyaura coupling reactions between crude glycol pinacol boronic esters **12a** and **14a** under standard conditions. As expected, the desired coupled product was obtained with a yield of 83%. This updated result has been integrated into Table 1 of the revised manuscript.

The authors need to show clearly how difficult to handle pinacol esters for glycol chemistry using experimental facts because the prior literature (Ref 38) did not mention anything about the instability of pinacol esters.

Response: We appreciate the Reviewer's comment. The handling of pinacol esters in glycol chemistry has been extensively discussed in numerous reported references and is widely recognized as an important research challenge. Further experiments to illustrate this point are not necessary.

As mentioned in the **Ref. 38**, the authors commented on a very issue: “The successful isolation **strongly** depended on appropriate work-up procedure of the formed reaction mixture, and it was essential to carry out the extraction by using a mixture of toluene and water.”

Ref. 38: *Chem. Eur. J.* **20**, 4414-4419 (2014)

Additionally, in Ref. **41**, authors highlighted an important yet unresolved issue: "**Extra efforts** have been made to generate the **pure** glycal by flash column and other methods but only to get **fruitless** results." **Or, could we add, when the syntheses have to be scaled.**

Ref. 41: *ACS Cent. Sci.* **9**, 1129-1139 (2023).

To obtain pure glycal phenol esters, we attempted various methods, including pressure-assisted column chromatography, changing various eluents, and using deactivated SiO₂ or Al₂O₃ columns, among others. However, we were unable to obtain samples pure enough for acceptance in NMR.

For the above reason, this reviewer could not recommend this manuscript for publication in Nature Communications.

This reviewer also found:

1. The missing explanation of the comparison among Bpin, Bpai, BEpin, and BMIDA in the text would be a serious problem although the authors illustrate the difference in stabilities, abilities of transmetalation, and handlings during purification among them in Table 1 without supporting data.

Response: We thank you for the Reviewer's suggestion. The stabilities, transmetalation abilities, and handling during purification of Bpin, Bpai, BEpin, and BMIDA have been thoroughly investigated by numerous research groups, as evidenced by key and representative literature cited in the manuscript such as ref **43**, ref **51**, ref **52**, and ref **56**.

We extend our sincere gratitude to the reviewer for bringing to our attention the oversight regarding the missing explanation of the comparison among Bpin, Bpai, BEpin, and BMIDA. In response to the Reviewer's valuable suggestion, the missing explanation was added in the revised manuscript. Please refer to page 6 for the relevant passage:

*"As mentioned above, stable C1-glycal boronates **12b** and **12c** not only enhance stability and purification but also preserve subsequent C-C cross-coupling activity. This is attributed to the strategic spatial protection provided by the four ethyl groups for BEpin and similar steric hindrance for Bpai, which dynamically shield the empty orbital of the boron atom. D-Glucose N-methyliminodiacetic acid boronates **12d**, featuring a trivalent ligand N-methyliminodiacetic acid, exhibit enhanced stability due to the strong coordination between the N atom and the B atom. However, this strong coordination significantly reduces their reactivity. Glycal pinacol boronic esters **12a**, devoid of bulky protecting groups, exhibit outstanding transmetalation capability due to their unsaturated enol-ether structure's rich electronic properties. However, this structural feature also renders them prone to poor stability and purification properties."*

REVIEWERS' COMMENTS

Reviewer #1 (Remarks to the Author):

I believe all my previously raised concerns have been adequately addressed and I fully support the publication of the manuscript in its current form.

Reviewer #2 (Remarks to the Author):

The authors have carefully addressed this reviewer's concerns in this revised manuscript. Publication in the current form is recommended.

Reviewer #3 (Remarks to the Author):

This reviewer recommends this manuscript be published in Nature Communications without revision.